# Solving Time-Dependent Differential Equations with Physical Dynamical Systems

**Chuan Liu** [1]   **Yijie Chen** [1]   **Ruibing Song** [1]   **Wenhao Huang** [1]   **Chunshu Wu** [2]   **Deqian Kong** [3]   **Ying Nian Wu** [3]
**Kaiyuan Yang** [1]   **Ang Li** [4]   **Tony (Tong) Geng** [1]

## Abstract

Time-Dependent Differential Equations (TDDEs) model dynamical processes across science and engineering, but time-critical applications require solvers that deliver high-fidelity trajectories under stringent latency constraints. Most existing TDDE solvers are limited by time discretization, forcing a latency-accuracy trade-off where smaller step sizes capture high-fidelity trajectories but incur prohibitive runtime, while larger steps meet real-time budgets at the cost of trajectory distortion. Dynamical System Machines (DSMs) offer a promising alternative by computing through continuous physical evolution, yet existing DSMs struggle to capture the spatiotemporal complexity of TDDEs. This work introduces DS-TS, a novel TDDE solver that is both accurate and efficient by leveraging the unique computational advantages of DSMs. DS-TS integrates three key innovations: (1) Excitatory-Inhibitory Inspired Coupling to better model complex spatial interactions; (2) State-aware Dynamic Nonlinearity to enable rich inter-node interactions and state-dependent spatiotemporal correlations; and (3) Hierarchical Temporal Integration to capture high-order temporal dependencies. Experiments demonstrate that DS-TS achieves high-fidelity solutions while delivering orders-of-magnitude improvements in speed ($\sim 10^3 \times$) and energy efficiency ($\sim 10^5 \times$) compared to baseline solvers.

[1]Department of Electrical and Computer Engineering, Rice University, Houston, TX, USA [2]Pacific Northwest National Laboratory, Richland, WA, USA [3]Department of Statistics and Data Science, University of California Los Angeles, Los Angeles, CA, USA [4]Department of Electrical and Computer Engineering, University of Washington, Seattle, WA, USA. Correspondence to: Tony (Tong) Geng <tg62@rice.edu>.

*Proceedings of the $43^{rd}$ International Conference on Machine Learning*, Seoul, South Korea. PMLR 306, 2026. Copyright 2026 by the author(s).

## 1. Introduction

Time-Dependent Differential Equations (TDDEs) are the mathematical backbone for describing how physical, chemical, and biological systems evolve (Ascher et al., 1995; Regazzoni et al., 2019). Classic examples include reaction-diffusion equations in chemical and biological systems (Pinar, 2021; Hariharan & Kannan, 2014), wave equations describing acoustic, elastic, and electromagnetic propagation (Ishimaru, 2017; Lannes & Bonneton, 2009), and advection-diffusion equations that capture transport and mixing in fluids and porous media (Dehghan, 2004; Demirdjian et al., 2022). The pursuit of accurate and efficient TDDE solvers has continually advanced scientific discovery and engineering innovation. Recently, the demand for real-time TDDE computation has become pressing as solvers are increasingly embedded in time-critical settings such as real-time feedback control, in-the-loop simulation, and interactive scientific computing, where models must deliver results under stringent latency constraints, often at microsecond or nanosecond scales (Chen et al., 2025).

Existing approaches struggle to meet this goal due to a fundamental limitation introduced by time discretization (Regazzoni et al., 2019; Tadmor, 2012; Kumar & Yadav, 2025). Most solvers depend on discrete time stepping, creating an inherent trade-off in latency-constrained settings (Chen et al., 2025). For numerical solvers, achieving sufficient temporal resolution for high-fidelity trajectories can substantially increase computational cost, whereas coarser stepping to satisfy real-time constraints degrades trajectory accuracy (Tadmor, 2012). While machine learning (ML) surrogates can compress long-horizon evolution into a single forward evaluation and thereby offer speedups, they are often trained on solver-generated trajectories at discrete time steps and consequently still balance efficiency against faithful tracking of continuous-time trajectories (Lu et al., 2021b; Takamoto et al., 2022). These limitations motivate the search for approaches that preserve continuous-time, high-fidelity TDDE solutions while operating efficiently under real-time constraints.

We find that recently emerging Dynamical System Machines (DSMs) (Afoakwa et al., 2021; Wu et al., 2024;

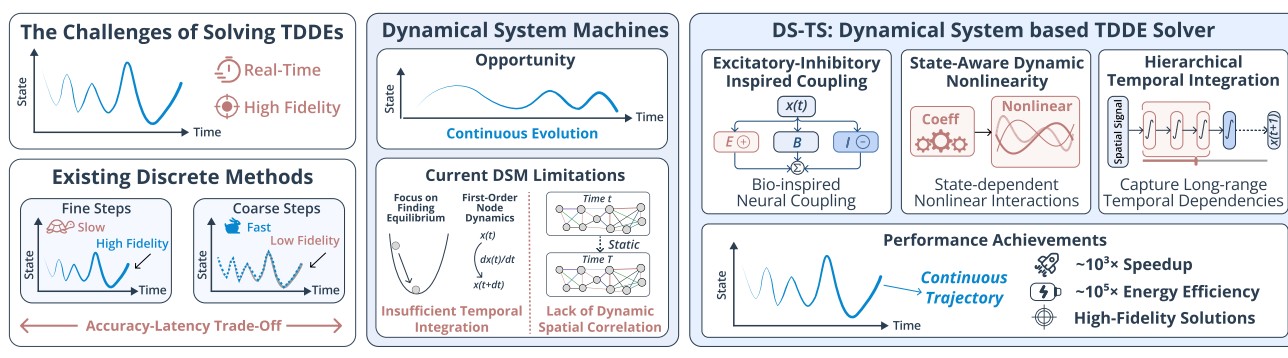

*Figure 1.* Overview of DS-TS. *Left:* Existing TDDE solvers suffer from a trade-off. *Middle:* Existing dynamical system machines enable continuous evolution but remain limited. *Right:* DS-TS bridges these gaps with three key design components.

Mohseni et al., 2022; Liu et al., 2025a;c) offer a promising path toward real-time TDDE solving. DSMs comprise programmable electronic components that physically embody dynamical systems, such that computation is realized through the machine's intrinsic continuous evolution, which can often be modeled by differential equations. As a result, the physical evolution of DSMs provides a native and efficient substrate for solving TDDEs in continuous time.

Despite their promise, current DSMs face critical limitations that prevent them from realizing their potential for solving TDDEs. *First, insufficient temporal integration capability.* Existing DSMs (Afoakwa et al., 2021; Wu et al., 2024; 2025; Liu et al., 2025a;c) focus on finding equilibrium states, failing to capture the evolving trajectories of systems over time, which is the defining characteristic of TDDEs. Moreover, many TDDEs involve high-order time derivatives (e.g., acceleration), while existing DSMs are limited to first-order dynamics, rendering them incapable of capturing these high-order temporal dependencies essential for accurate TDDE solving. *Second, inability to handle dynamic spatial correlations.* When temporal evolution is considered, the spatial correlations among nodes can become dynamic and increasingly intricate. This necessitates adaptive spatial correlation mechanisms capable of tracking time-varying dependencies. However, existing DSMs (Afoakwa et al., 2021; Wu et al., 2024; 2025; Liu et al., 2025a;c) employ static spatial representations and fail to capture these dynamic correlations.

As shown in Fig. 1, this work addresses these challenges by introducing *DS-TS*, a novel ultra-efficient and accurate TDDE solver that leverages the unique computational power of DSMs through three synergistic innovations. ① *Excitatory-Inhibitory Inspired Coupling*: Drawing inspiration from biological neural networks where excitatory and inhibitory neurons jointly form system dynamics, we design a new spatial coupling mechanism that augments spatial interactions with complementary excitatory and inhibitory dynamics, thereby enhancing its expressive capacity for modeling complex spatial interactions. ② *State-aware Dy-*

*namic Nonlinearity*: We further introduce a state-aware dynamic nonlinearity expansion, which enriches inter-node interactions and introduces time-varying spatial correlations. Through continuous temporal evolution, this mechanism yields a vastly expanded effective parameter space, delivering superior expressivity for capturing complex spatiotemporal dynamics. ③ *Hierarchical Temporal Integration*: We extend standard DSM nodes with a configurable integration mechanism that continuously accommodates multiple orders of time derivatives. This enables each node to inherently capture high-order temporal dependencies, which are essential for accurately solving TDDEs with high-order time-derivative terms. Our key contributions are summarized as follows:

- We propose DS-TS, a novel TDDE solver that leverages the alignment between continuous physical evolution and TDDE solving to achieve highly efficient and accurate solutions.
- We integrate Excitatory-Inhibitory Inspired Coupling, State-aware Dynamic Nonlinearity, and Hierarchical Temporal Integration into a unified DS-TS design that addresses the intrinsic challenges of TDDE solving.
- Experiments demonstrate that DS-TS consistently achieves high-fidelity solutions while delivering orders-of-magnitude improvements in speed ($\sim 10^3 \times$) and energy efficiency ($\sim 10^5 \times$) compared to existing solvers.

## 2. Preliminaries

This section briefly introduces essential preliminaries for TDDEs and DSMs.

**Time-Dependent Differential Equations** form the foundation for modeling dynamic processes that evolve over time in physics, engineering, and applied sciences. Let $\Omega \subset \mathbb{R}^m$ denote the spatial domain, and let $t \in [0, T]$ denote the time interval. TDDEs can be expressed abstractly as:

$$\mathcal{F}\left(t, \mathbf{x}, u, \frac{\partial u}{\partial t}, \nabla_{\mathbf{x}} u, \nabla_{\mathbf{x}}^2 u, \dots\right) = 0, \qquad (1)$$

where $\mathbf{x} = (x_1, \dots, x_m) \in \Omega$, $u = u(t, \mathbf{x})$ is the unknown

solution, and $\mathcal{F}(\cdot)$ is a function that encodes the dynamics of the underlying system.

**Dynamical System Machines**
can be viewed as continuous dynamical systems whose $N$ internal node states evolve through specified interactions. Denote the node states by $\mathbf{x}(t) = [x_1(t), \ldots, x_N(t)]^\top$, where $x_i(t)$ represents the instantaneous state of node $i$ at time $t$. Computation is performed by designing the differential equation that drives $\mathbf{x}(t)$. For instance, the node dynamics can be designed as:

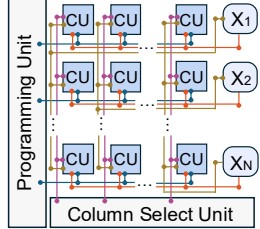

*Figure 2.* General architecture of DSMs.

$$\frac{dx_i}{dt} = \frac{1}{C} \left( \sum_{j \neq i}^{N} J_{ij} x_j - 2h_i x_i \right), \qquad (2)$$

where $J_{ij}$ parametrizes pairwise correlations between nodes $i$ and $j$, and $h_i$ controls a self-feedback strength. To physically build such a DSM, each node $x_i$ is represented as a voltage on a capacitor $C$, while parameters $\mathbf{J}$ and $\mathbf{h}$ are implemented as resistor conductances, forming a resistive coupling network as shown in Fig. 2 (CU refers to Coupling Units $\mathbf{J}$). Using Kirchhoff's current law at each capacitor node, computation is physically carried out through charging/discharging capacitor voltages at the "speed of electrons." By modifying the parameters (e.g., changing conductances $\mathbf{J}$ and $\mathbf{h}$) or redefining the node dynamics, DSMs can be reconfigured to continuously execute different computations and perform various tasks.

## 3. Methodology

In this section, we present the detailed design of DS-TS. We first introduce the key components of DS-TS: the Excitatory-Inhibitory Inspired Coupling (EIC, Section 3.1.1), the State-aware Dynamic Nonlinearity (SDN, Section 3.1.2), and the Hierarchical Temporal Integration (HTI, Section 3.1.3). We then describe how to align the dynamics of DS-TS with a target TDDE in Section 3.2, followed by the handling of boundary conditions in Section 3.3. Finally, we introduce the physical implementation of DS-TS in Section 3.4.

### 3.1. DS-TS Design

#### 3.1.1. EXCITATORY-INHIBITORY INSPIRED COUPLING

We consider a TDDE discretized over $N$ spatial nodes, with each grid point mapped to a node of DS-TS. Let $x_i(t)$ denote the state at node $i$, and let $\mathbf{x}(t) \in \mathbb{R}^N$ denote the global state vector. In computational neuroscience, the computation of cortical microcircuits is often described through

balanced excitatory-inhibitory (E-I) interactions: excitatory populations promote coordinated amplification, while inhibitory populations provide gain control, competition, and stabilization (Shipston-Sharman et al., 2016; Sadeh & Clopath, 2021). Motivated by this "push-pull" mechanism, we propose *Excitatory-Inhibitory Inspired Coupling* (EIC), a structured augmentation of spatial coupling that enhances expressivity while preserving stable linear mixing.

As shown in the left panel of Fig. 3, we start from a standard linear spatial signal:

$$B_i(t) = \sum_{j=1}^{N} W_{ij}^B x_j(t), \qquad (3)$$

where $W^B$ serves as a learnable weight matrix that provides backbone mixing. Then, we further introduce excitatory and inhibitory signals:

$$E_i(t) = \phi \left( \sum_{j=1}^{N} W_{ij}^E x_j(t) \right), I_i(t) = \phi \left( \sum_{j=1}^{N} W_{ij}^I x_j(t) \right), \quad (4)$$

with learnable weight matrices $W^E, W^I$ and a non-negative function $\phi(\cdot)$. The resulting EIC signal for node $i$ is:

$$B_i(t) + \alpha \left( E_i(t) - I_i(t) \right), \qquad (5)$$

where $\alpha$ controls the strength of the excitatory-inhibitory signals. The excitatory pathway $\mathbf{E}(t)$ captures cooperative interactions that amplify aligned spatial patterns, whereas the inhibitory pathway $-\mathbf{I}(t)$ induces competition that suppresses conflicting activity and controls gain. By combining them, EIC implements a biologically inspired push-pull modulation of spatial information flow, moving the couplings beyond purely linear mixing and improving the system's capacity to represent complex spatial dependencies.

#### 3.1.2. STATE-AWARE DYNAMIC NONLINEARITY

To capture high-order, dynamic nonlinear spatial interactions, DS-TS introduces *State-aware Dynamic Nonlinearity* (SDN). Polynomial expansions offer a standard tool for universal function approximation, yet conventional constructions employ fixed coefficients, rendering the induced nonlinearity static. SDN addresses this limitation by conditioning the polynomial coefficients on the current system state, allowing them to evolve along the trajectory and yielding a dynamic and expressive nonlinearity.

As shown in the middle panel of Fig. 3, SDN produces node-wise coefficients via a learned *coefficient generator*:

$$c_{m,i}(t) = \sum_{j=1}^{N} \left( W_m^C \right)_{ij} x_j(t), \quad m = 0, \ldots, D, \quad (6)$$

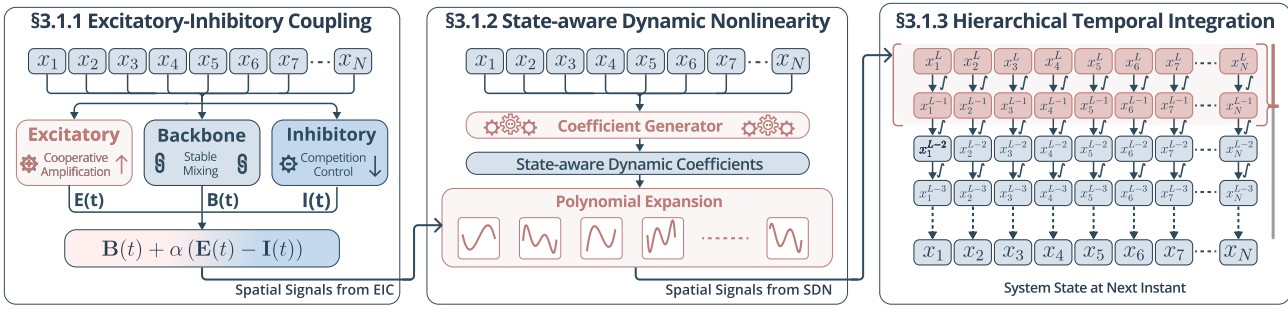

*Figure 3.* DS-TS Design. *Left:* Excitatory-Inhibitory Inspired Coupling incorporates excitatory and inhibitory pathways to better model spatial interactions. *Middle:* State-aware Dynamic Nonlinearity generates state-dependent coefficients to model dynamic, nonlinear node interactions. *Right:* Hierarchical Temporal Integration constructs a chain of integrators to capture high-order temporal dependencies.

where $W_m^C$ are trainable parameters, and $m$ is the polynomial degree. This construction makes each $c_{m,i}(t)$ a function of the instantaneous system state $\mathbf{x}(t)$, thereby allowing the local nonlinearity to adjust as the system evolves. Combined with the signals from the EIC module, the resulting SDN node dynamics are:

$$\frac{dx_i(t)}{dt} = \sum_{m=0}^{D} c_{m,i}(t) \left( B_i(t) + \alpha \left( E_i(t) - I_i(t) \right) \right)^m,$$

$$(7)$$

for $i = 1, 2, \ldots, N$. Because $c_{m,i}(t)$ is computed from the instantaneous state at every time $t$, the effective nonlinear operator varies along the trajectory, enabling DS-TS to model dynamic, nonlinear spatiotemporal interactions.

### 3.1.3. HIERARCHICAL TEMPORAL INTEGRATION

Although the proposed EIC and SDN effectively capture complex inter-node interactions, many TDDEs further require modeling dynamics governed by high-order time derivatives. In contrast, existing DSMs are typically formulated as first-order systems, which limits their capacity to represent such dynamics. To address this limitation, DS-TS introduces *Hierarchical Temporal Integration* (HTI), which augments each node with a hierarchy of augmented states to faithfully represent intricate high-order temporal behaviors.

As shown in the right panel of Fig. 3, for node $i$, we define its augmented temporal state as:

$$\tilde{\mathbf{x}}_i(t) = \left[ x_i^{(0)}(t), x_i^{(1)}(t), \ldots, x_i^{(L)}(t) \right]^\top, \quad (8)$$

where $x_i^{(0)}(t)$ denotes the TDDE solution at node $i$ and time $t$, and $\{x_i^{(\ell)}(t)\}_{\ell=1}^{L}$ serve as additional temporal states that encode higher-order temporal behaviors. HTI ties these states through the following hierarchical chain:

$$\frac{dx_i^{(\ell)}(t)}{dt} = x_i^{(\ell+1)}(t), \qquad \ell = 0, 1, \ldots, L-1. \quad (9)$$

Here, $L$ denotes the integration depth, and $x_i^{(\ell)}(t)$ represents the $\ell$-th temporal state at node $i$.

The highest-level state $x_i^{(L)}(t)$ is driven by the effective signal obtained by applying SDN to the base-layer states $\mathbf{x}^{(0)}(t) = \left[ x_1^{(0)}(t), \ldots, x_N^{(0)}(t) \right]^\top$:

$$\frac{dx_i^{(L)}(t)}{dt} = \sum_{m=0}^{D} c_{m,i}(t) \left( B_i(t) + \alpha \left( E_i(t) - I_i(t) \right) \right)^m.$$

$$(10)$$

Here, $B_i(t)$, $E_i(t)$, and $I_i(t)$ follow Eqs. (3)-(4) applied to $\mathbf{x}^{(0)}(t)$. HTI establishes a hierarchy of coupled integrators, with the SDN-induced signal injected at the highest temporal level and propagated through the integration chain. By doing so, it yields a first-order augmented representation of high-order temporal dynamics, enabling DS-TS to effectively model the temporal complexity of the target TDDEs.

### 3.2. Hardware-Aware Training

DS-TS addresses a target TDDE by decomposing its spatiotemporal complexity and distributing each part to the corresponding component. Hierarchical Temporal Integration (HTI) handles temporal integration by activating an appropriate number of auxiliary temporal states for each node, thereby matching the order of the time derivatives in the target TDDE. The decomposed spatial complexity in the highest-order temporal layer $L$ is then handled by EIC and SDN, which parameterize nonlinear and state-dependent node interactions. Let $\mathbf{x} \in \mathbb{R}^N$ denote the state vector at layer $L$, and let $\mathcal{S}^\star : \mathbb{R}^N \to \mathbb{R}^N$ denote the corresponding ground-truth operator. Given a collection of sampled states $\{\mathbf{x}^{(h)}\}_{h=1}^{H}$, we compute targets $\mathbf{s}^{(h)} := \mathcal{S}^\star(\mathbf{x}^{(h)})$, yielding a supervised dataset $\mathcal{D} = \{(\mathbf{x}^{(h)}, \mathbf{s}^{(h)})\}_{h=1}^{H}$. Since DS-TS also defines a learnable operator $\widehat{\mathcal{S}}_\Theta(\mathbf{x})$ that corresponds to Eq. (10), we can optimize DS-TS's parameters $\Theta$ via:

$$\min_\Theta \mathcal{L}_{\text{MSE}} = \frac{1}{H} \sum_{h=1}^{H} \left\| \widehat{\mathcal{S}}_\Theta \left( \mathbf{x}^{(h)} \right) - \mathbf{s}^{(h)} \right\|_2^2, \quad (11)$$

where $\Theta = \{W^B, W^E, W^I, W^C, \alpha\}$.

To ensure reliable deployment on DSM hardware, we adopt *Quantization-aware Training* (QAT) to account for the limited bit precision of resistor conductances. In practice, we

equip the parameter set $\Theta$ with QAT settings and integrate fake-quantization observers into the training pipeline. During optimization, the forward pass emulates the target 8-bit integer representation, whereas the backward pass operates on full-precision floating-point values. This strategy enables the optimizer to align parameters with discrete quantization levels and improves robustness to quantization-induced perturbations. Upon convergence, we finalize the model in its quantized configuration, thereby ensuring that the DSM accurately realizes the trained dynamics. More details are provided in Appendix A.2.

### 3.3. Boundary Condition Enforced Inference

With optimized parameters, DS-TS performs inference by continuously and jointly evolving all components to generate trajectories of the target TDDE. Another important advantage of DS-TS is its node-wise specification of the dynamics, which provides a natural and explicit way to enforce boundary conditions (BCs). Unlike many ML-based solvers that require auxiliary constraints, penalty terms, or post-processing procedures to impose BCs, DS-TS incorporates them directly into the continuous evolution by prescribing suitable dynamics at boundary nodes. As a result, the BCs are naturally maintained throughout the integrated trajectory. Specifically, we partition the set of $N$ nodes into an interior node set $\mathcal{I}$ and a boundary node set $\mathcal{B}$, such that $\mathcal{I} \cup \mathcal{B} = \{1, \ldots, N\}$ and $\mathcal{I} \cap \mathcal{B} = \emptyset$.

*Interior node dynamics.* Following the design in Sec. 3.1, the dynamics of interior nodes $i \in \mathcal{I}$ are specified by:

$$\frac{dx_i^{(L)}(t)}{dt} = \sum_{m=0}^{D} c_{m,i}(t)\left(B_i(t) + \alpha\left(E_i(t) - I_i(t)\right)\right)^m,$$

(12)

$$\frac{dx_i^{(\ell)}(t)}{dt} = x_i^{(\ell+1)}(t), \quad \ell = 0, 1, \ldots, L-1,$$

(13)

where $\mathcal{I}$ denotes the set of interior nodes. $B_i(t)$, $E_i(t)$, and $I_i(t)$ follow Eqs. (3)-(4) applied to $\mathbf{x}^{(0)}(t)$.

*Boundary node dynamics.* For boundary nodes $i \in \mathcal{B}$, DS-TS configures them according to the prescribed BCs. Consider the commonly used *Periodic BCs*, where boundary nodes at opposite ends are topologically identified. Let $\pi$ be the pairing map (e.g., leftmost $\leftrightarrow$ rightmost). We enforce:

$$x_i^{(\ell)}(t) = x_{\pi(i)}^{(\ell)}(t), \quad \ell = 0, \ldots, L,$$

(14)

by tying the corresponding parameters and initial conditions. Specifically, the DS-TS parameters are configured so that each pair shares the same state and receives identical signals from their neighbors, thereby maintaining periodic boundary consistency throughout the evolution. For general time-dependent BCs that prescribe boundary values $g_i(t)$ on the

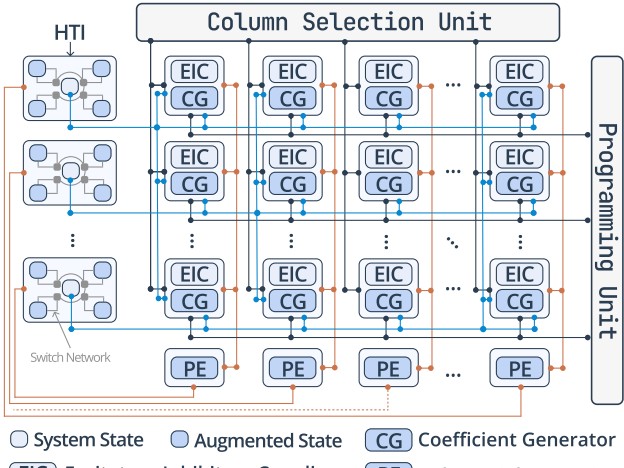

| Column Selection Unit |

System State ○  Augmented State ○  Coefficient Generator CG
EIC Excitatory-Inhibitory Coupling  PE Polynomial Expansion

*Figure 4.* The physical implementation of DS-TS.

boundary set $\mathcal{B}$, we treat boundary states as known signals. Specifically, for each boundary node $i \in \mathcal{B}$, we impose:

$$x_i^{(0)}(t) = g_i(t), \quad x_i^{(\ell)}(t) = \frac{d^\ell g_i(t)}{dt^\ell},$$

(15)

where $\ell = 1, \ldots, L$. In this way, $x_i^{(0)}(t) = g_i(t)$ corresponds to the prescribed boundary value, while the higher-level states are set consistently with its successive temporal derivatives. This mechanism enforces the time-dependent BCs directly throughout the continuous evolution.

Overall, the key insight is that DS-TS treats boundary conditions as intrinsic components of node dynamics, rather than as external constraints. This simplifies implementation while enforcing physical consistency, yielding more accurate and stable solutions.

### 3.4. Physical Implementation of DS-TS

As shown in Fig. 4, the proposed DS-TS is implemented as a direct extension of prior DSMs, preserving the same circuit-level representations. As in existing DSM implementations, node states are represented by capacitor voltages, whereas learned interaction weights, including $W^B$, $W^E$, and $W^I$, are implemented using programmable conductance arrays, with signed weights realized through differential conductance pairs. The trained scalar $\alpha$ is absorbed into $W^E$ and $W^I$ prior to hardware mapping, so its effect is encoded in resistor conductances. Under this mapping, $B_i(t)$, $E_i(t)$, and $I_i(t)$ are naturally realized as electrical currents. To realize the branches $E_i(t)$ and $-I_i(t)$, we use sign-gating with a voltage-sensed comparator and two selectable paths: the input current $I_{\text{in}}$ is converted to a sensing voltage $V_{\text{sense}}$ and compared against a reference $V_{\text{ref}} = 0$. The $E_i(t)$ branch forwards the signal when $V_{\text{sense}} \geq V_{\text{ref}}$, while the $-I_i(t)$ branch forwards the signal when $V_{\text{sense}} \leq V_{\text{ref}}$.

The SDN is likewise constructed from DSM-compatible

primitives. The coefficient-generator parameters $W_m^C$ are realized using programmable conductance arrays, enabling the coefficient signals $c_{m,i}(t)$ to be generated as electrical currents. The currents produced by the upstream EIC block are summed according to Kirchhoff's Current Law (KCL) to form the effective SDN input current $I_{\text{IN}}(t) = B_i(t) + \alpha \left( E_i(t) - I_i(t) \right)$. The SDN then realizes the polynomial mapping $I_{\text{OUT}}(t) = \sum_{m=0}^{D} c_{m,i}(t) \left( I_{\text{IN}}(t) \right)^m$. Since the current may be positive or negative, corresponding to two opposite current directions, a current comparator first detects its polarity and routes it to one of two complementary branches: an NMOS-based current-sinking branch for positive current flowing into the node, and a PMOS-based current-sourcing branch for negative currents flowing out of the node. Then, a subsequent logarithmic-domain computation pipeline operates on the selected branch, which includes a current conveyor, a log transformer (LT), a voltage-scaling stage, and an exponential transformer (ET). The current conveyor isolates the upstream node and maintains the EIC output at a stable voltage. For polynomial orders $m \geq 1$, the LT converts $I_{\text{IN}}(t)$ to a voltage $V_{\text{LT}}(t) \propto \ln \left( I_{\text{IN}}(t) \right)$. In parallel, each coefficient current $c_{m,i}(t)$ is also converted to a voltage $V_{c_{m,i}}(t) \propto \ln \left( c_{m,i}(t) \right)$. The scaling stage then generates $m V_{\text{LT}}(t) + V_{c_{m,i}}(t)$. The ET subsequently maps this summed voltage back to the current domain, yielding $c_{m,i}(t) \left( I_{\text{IN}}(t) \right)^m$. Finally, the $D+1$ branch currents, corresponding to polynomial degrees ($m = 0, \ldots, D$), are summed according to KCL to produce $I_{\text{OUT}}(t)$.

Similarly, for HTI, the augmented temporal states are also implemented as capacitor voltages. The system node and its augmented temporal states are connected through a programmable switch network, allowing the temporal integration depth to be configured in hardware. Each temporal state is realized by a capacitor and a transconductance ($G_m$) cell. Consistent with the HTI relation $\frac{dx_i^{(\ell)}(t)}{dt} = x_i^{(\ell+1)}(t)$, the voltage corresponding to $x_i^{(\ell+1)}(t)$ is converted by the $G_m$ cell into a current that charges or discharges the capacitor storing $x_i^{(\ell)}(t)$. This forms a configurable chain of coupled integrators that physically realizes the hierarchical temporal integration required by HTI.

## 4. Evaluation

### 4.1. Experimental Setup

**Datasets.** We evaluate on five representative TDDE benchmarks that span diverse physical phenomena: Advection-Diffusion (Wang & Wang, 2011), Wave (Moseley et al., 2020), Reaction-Diffusion (Takamoto et al., 2022), Fokker-Planck (Risken, 1989), and a high-order Hyperbolic equation (Shu, 2016). These problems exhibit markedly different temporal behaviors and solution characteristics, ranging from transport-dominated dynamics to wave-like propaga-

tion and nonlinear reaction kinetics. The selection includes both parabolic and hyperbolic equations, as well as systems with first-order and higher-order time derivatives, forming a rigorous and diverse evaluation suite. Full dataset specifications are provided in Appendix A.1.

**Baselines.** We follow the evaluation protocol of (Takamoto et al., 2022) and include baselines from different families: (1) *Physics-Informed Neural Networks (PINNs)* (Raissi et al., 2019); (2) *Fourier Neural Operator (FNO)* (Li et al., 2021); (3) *UNet* (Takamoto et al., 2022); (4) existing DSMs, specifically *NP-GL* (Wu et al., 2024), *DS-TPU* (Wu et al., 2025), and *EADS* (Liu et al., 2025c). For FNO and UNet, we adopt the commonly used autoregressive training strategy for trajectory generation under two context lengths: AR-1 (predict $u_{t+1}$ from $u_t$) and AR-16 (predict $u_{t+1}$ from $\{u_{t-15}, \ldots, u_t\}$). All baseline implementations use their originally published settings to ensure fair comparison. To evaluate performance at different time horizons, we report: (1) short-term results via 100-step free rollouts (ST); and (2) long-term results via 500-step free rollouts (LT). The short-term scenario evaluates immediate prediction accuracy and basic properties, while the long-term scenario stress-tests stability, error accumulation, and the tendency toward instability over extended periods. We additionally evaluate three spatial discretizations: R1 (256 grids), R2 (512 grids), and R3 (1024 grids), to assess whether the solvers maintain accuracy across different spatial discretizations.

**Experimental Platforms.** For conventional ML-based solvers, including PINN, FNO, and UNet, we run inference on an NVIDIA A100 40GB SXM GPU and report both accuracy and per-step GPU latency. For prior DSMs, including NP-GL, DS-TPU, and EADS, we reproduce their original software simulators following the corresponding implementation settings, and report the accuracy and per-step latency. DS-TS is evaluated with a mixed evaluation flow that combines software simulation and hardware characterization. Direct full-system circuit simulation of a large-scale DS-TS instance is impractical, and full-chip Cadence runs would be prohibitively slow for the problem sizes considered in our experiments. Therefore, we simulate the global state evolution in a custom software simulator, while extracting accurate behavior for electrically sensitive components (e.g., SDN) from circuit-level characterization. More implementation details of DS-TS are provided in Appendix A.2. The power of DS-TS is estimated using the Cadence Mixed-Signal Design Environment under a 180 nm CMOS process. For direct comparison with existing DSMs, we report DS-TS power to a reference scale of 2000 nodes, yielding 1.12 W. At this scale, the reported power for NP-GL, DS-TPU, and EADS is 260 mW, 1.6 W, and 1.2 W, respectively. As additional reference points for GPU-based solvers, the NVIDIA A100 GPU is rated at 250 W.

*Table 1.* MAE comparison of DS-TS and baselines on selected TDDEs. The best results are highlighted in bold.

| Method | Advection-Diffusion | | | Wave | | | Reaction-Diffusion | | | Fokker-Planck | | | Hyperbolic | | |
|---|---|---|---|---|---|---|---|---|---|---|---|---|---|---|---|
| | R1 | R2 | R3 | R1 | R2 | R3 | R1 | R2 | R3 | R1 | R2 | R3 | R1 | R2 | R3 |
| **Short-Term Scenarios** | | | | | | | | | | | | | | | |
| PINN | 1.21e-4 | 1.53e-4 | 1.75e-4 | 9.10e-4 | 9.38e-4 | 9.72e-4 | 1.26e-4 | 4.32e-4 | 9.28e-4 | 3.46e-4 | 3.22e-4 | 3.11e-4 | 2.79e-4 | 2.74e-4 | 4.83e-4 |
| UNet (AR-1) | 6.55e-3 | 9.22e-3 | 8.38e-3 | 1.56e-2 | 1.67e-2 | 2.35e-2 | 1.24e-2 | 3.07e-2 | 4.82e-2 | 1.14e-2 | 2.14e-2 | 3.28e-2 | 1.72e-2 | 2.62e-2 | 3.95e-2 |
| UNet (AR-16) | 2.33e-3 | 1.30e-3 | 2.69e-3 | 1.65e-3 | 1.89e-3 | 7.41e-3 | 4.11e-3 | 5.62e-3 | 7.67e-3 | 1.03e-3 | 1.26e-3 | 2.61e-3 | 1.10e-3 | 1.81e-3 | 2.66e-3 |
| FNO (AR-1) | 1.02e-4 | 2.63e-4 | 8.31e-4 | 3.85e-3 | 5.94e-3 | 6.47e-3 | 1.52e-3 | 2.09e-3 | 6.01e-3 | 2.25e-3 | 3.81e-3 | 4.92e-3 | 1.79e-3 | 2.55e-3 | 1.46e-3 |
| FNO (AR-16) | 1.72e-4 | 2.68e-4 | 1.11e-4 | 1.12e-4 | 1.08e-4 | 9.42e-5 | 1.98e-4 | 3.00e-4 | 3.69e-4 | 2.41e-4 | 1.43e-4 | 5.56e-4 | 2.04e-4 | 3.65e-4 | 1.41e-4 |
| NP-GL | 5.37e-3 | 6.80e-3 | 7.78e-3 | 6.15e-2 | 6.38e-2 | 8.07e-2 | 5.67e-3 | 7.43e-3 | 8.19e-3 | 3.18e-2 | 3.35e-2 | 7.99e-2 | 1.10e-2 | 1.85e-2 | 3.31e-2 |
| DS-TPU | 3.12e-3 | 4.71e-3 | 5.39e-3 | 3.06e-2 | 3.15e-2 | 4.82e-2 | 4.19e-3 | 4.86e-3 | 6.52e-3 | 5.97e-3 | 5.51e-3 | 6.84e-3 | 7.52e-3 | 8.16e-3 | 9.64e-3 |
| EADS | 2.91e-3 | 3.12e-3 | 3.95e-3 | 2.47e-3 | 3.72e-3 | 4.18e-3 | 2.68e-3 | 3.61e-3 | 4.16e-3 | 2.71e-3 | 3.62e-3 | 4.05e-2 | 3.73e-3 | 4.71e-3 | 4.95e-3 |
| DS-TS | **1.03e-6** | **1.37e-6** | **1.91e-6** | **1.71e-5** | **1.74e-5** | **1.76e-5** | **4.15e-6** | **7.12e-6** | **8.17e-6** | **2.23e-6** | **3.03e-6** | **3.64e-6** | **1.54e-6** | **1.25e-6** | **2.57e-6** |
| **Long-Term Scenarios** | | | | | | | | | | | | | | | |
| PINN | 1.49e-4 | 3.11e-4 | 4.35e-4 | 9.92e-4 | 1.01e-3 | 1.57e-3 | 3.31e-4 | 8.56e-4 | 4.01e-4 | 4.63e-4 | 4.28e-4 | 5.17e-4 | 2.91e-4 | 3.09e-4 | 5.41e-4 |
| UNet (AR-1) | 2.73e-2 | 3.31e-2 | 2.49e-2 | 4.58e-2 | 6.89e-2 | 8.64e-2 | 4.60e-2 | 6.48e-2 | 7.57e-2 | 4.50e-2 | 8.25e-2 | 9.32e-2 | 5.96e-2 | 7.96e-2 | 8.01e-2 |
| UNet (AR-16) | 1.74e-2 | 1.09e-2 | 2.01e-2 | 1.22e-2 | 1.47e-2 | 2.62e-2 | 1.12e-2 | 2.27e-2 | 4.45e-2 | 4.87e-3 | 5.93e-3 | 8.16e-3 | 3.75e-3 | 5.19e-3 | 7.72e-3 |
| FNO (AR-1) | 2.51e-3 | 1.46e-3 | 1.70e-3 | 2.98e-2 | 3.25e-2 | 6.26e-2 | 1.04e-2 | 1.62e-2 | 3.48e-2 | 1.63e-2 | 2.17e-2 | 3.23e-2 | 1.31e-2 | 3.74e-2 | 1.19e-2 |
| FNO (AR-16) | 1.33e-3 | 2.25e-3 | 4.49e-4 | 1.59e-3 | 1.32e-3 | 7.99e-4 | 1.15e-3 | 1.28e-3 | 1.46e-3 | 3.53e-3 | 2.02e-3 | 6.24e-3 | 1.16e-3 | 2.53e-3 | 8.51e-4 |
| NP-GL | 3.61e-2 | 4.15e-2 | 6.29e-2 | 3.34e-1 | 3.59e-1 | 6.47e-1 | 4.61e-2 | 6.19e-2 | 7.27e-2 | 1.18e-1 | 1.71e-1 | 4.57e-1 | 6.37e-2 | 7.98e-2 | 8.89e-2 |
| DS-TPU | 2.17e-2 | 2.81e-2 | 3.72e-2 | 7.92e-2 | 8.15e-2 | 9.53e-2 | 3.81e-2 | 4.63e-2 | 6.19e-2 | 3.26e-2 | 3.12e-2 | 4.24e-2 | 3.95e-2 | 4.20e-2 | 5.24e-2 |
| EADS | 1.83e-2 | 2.47e-2 | 2.91e-2 | 1.35e-2 | 1.59e-2 | 2.16e-2 | 2.72e-2 | 3.99e-2 | 5.41e-2 | 1.35e-2 | 2.29e-2 | 2.83e-2 | 1.26e-2 | 2.74e-2 | 2.91e-2 |
| DS-TS | **4.33e-6** | **5.19e-6** | **8.48e-6** | **4.99e-4** | **5.07e-4** | **5.14e-4** | **3.15e-5** | **3.22e-5** | **4.16e-5** | **8.96e-6** | **1.21e-5** | **1.40e-5** | **7.09e-6** | **1.02e-5** | **1.24e-5** |

*Figure 5.* Comparison of latency for DS-TS and baselines across the selected TDDEs.

## 4.2. Accuracy Evaluation

Table 1 reports mean absolute errors (MAE) in scientific notation (e.g., $e\text{-}6 = 10^{-6}$) for all evaluated methods on the selected TDDE benchmarks. DS-TS achieves the lowest MAE overall, demonstrating consistent and substantial improvements across the entire evaluation. Averaged across different benchmarks, DS-TS achieves an MAE that is one to three orders of magnitude lower than those of ML solvers (PINN, UNet, FNO), while also substantially improving upon prior DSMs. In short-term scenarios, DS-TS consistently achieves MAE values on the order of $10^{-6}$ to $10^{-5}$ across all benchmarks, while the best competing methods generally remain around $10^{-4}$. In long-term scenarios, DS-TS maintains the best performance with clear margins that often span one to two orders of magnitude, indicating strong temporal stability despite compounding rollout errors. These findings collectively show that DS-TS functions as an accurate and reliable TDDE solver: it maintains accuracy over extended rollouts, generalizes across different TDDE types, and successfully tackles high-order-in-time equations that previously challenged existing DSMs. This combination of stability and accuracy highlights DS-TS as an important advance and motivates its broader use in scientific computing and other applications.

## 4.3. Latency and Energy Evaluation

Fig. 5 compares the per-output-step latency across all benchmark equations, highlighting the substantial efficiency advantage of DS-TS. Since the evaluated methods operate under different computational paradigms, we define latency based on the time required to produce output samples at the same temporal points, enabling a fair comparison across methods. For discrete-time methods, latency is measured as the time required to perform one complete forward pass and generate the next-step output. For DS-TS, which evolves in continuous time rather than through discrete timesteps, latency is measured as the average temporal interval between consecutive queried samples, with samples taken at the same temporal points as the outputs produced by the discrete baselines. Under this matched sampling protocol, DS-TS achieves an approximate $10^3\times$ speedup over the ML-based solvers, while maintaining latency comparable to prior DSM implementations. The energy consumption in Fig. 6 shows an even larger advantage. Across all benchmark equations, GPU-based executions of the ML-based solvers incur the highest energy consumption, requiring approximately $10^5\times$ more energy than DS-TS. Overall, these results show that DS-TS substantially reduces both latency and energy consumption compared to ML-based solvers.

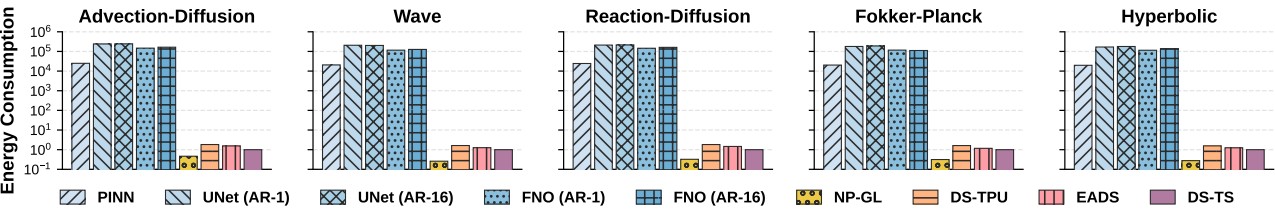

*Figure 6.* Comparison of energy consumption for DS-TS and baselines across the selected TDDEs.

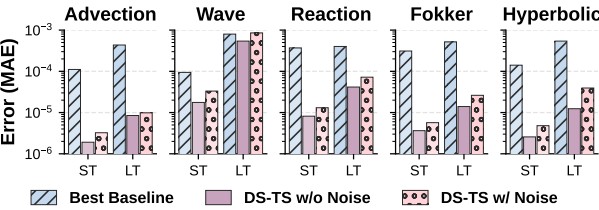

*Figure 7.* MAE comparison between DS-TS (w/o and w/ noise) and the lowest MAE among baselines on selected TDDEs in R3. "Advection" denotes "Advection-Diffusion," "Reaction" denotes "Reaction-Diffusion," and "Fokker" denotes "Fokker-Planck."

### 4.4. Robustness Evaluation

To quantify the impact of hardware noise on DS-TS, we model the dominant noise sources in our implementation, including resistor thermal noise and MOS transistor thermal noise. For each circuit block $k$ in DS-TS, we run Cadence simulations to extract the small-signal transfer function $H_i(f)$ together with the corresponding input-referred noise power spectral density $S_{n,i}(f)$. Aggregating the contributions of all blocks yields the overall noise as below:

$$\sigma = \left( \int \sum_{i=1}^{n} \left| \prod_{m=i}^{n} H_m(f) \right|^2 S_{n,i}(f)\,df \right)^{1/2}, \quad (16)$$

where $n$ denotes the number of cascaded blocks and $f$ denotes the effective signal bandwidth. We set $f = 1$ MHz, which covers our operating conditions. Under this setup, the simulated current-domain signal-to-noise ratio is 52 dB. Fig. 7 and Fig. 8 (Appendix A.3) report MAE under the above noise model. The resulting accuracy degradation is modest across all test cases: averaged over configurations, the MAE increases by less than a factor of two relative to the ideal noiseless DS-TS simulation. Importantly, DS-TS remains more accurate than the strongest baseline in every configuration. Overall, DS-TS maintains high accuracy while demonstrating strong robustness to noise.

## 5. Related Work

### 5.1. Numerical Solvers

Classical numerical solvers for TDDEs rely on spatial and temporal discretization. They typically employ the method of lines (Verwer & Sanz-Serna, 1984), discretizing the spa-tial domain via finite difference, finite element, or finite volume methods (Mazumder, 2015; Zlámal, 1968), thereby converting the TDDE into a system of ODEs that is subsequently integrated in time. Time integration typically uses explicit or implicit schemes. Explicit methods (e.g., forward Euler, Runge-Kutta (Hu & Wang, 2024)) are simple and parallelizable but demand small time steps for stability, substantially increasing computational cost (Hairer et al., 1993). Implicit schemes (e.g., backward Euler, Crank–Nicolson (Luskin et al., 1982)) relax stability constraints and better handle stiffness, but require solving large linear or nonlinear systems at each step, incurring significant computational and memory overhead (Ascher & Petzold, 1998). Techniques such as adaptive time-stepping schemes can alleviate these costs but do not remove them, thereby limiting flexibility and scalability in complex settings.

### 5.2. ML Solvers

ML solvers for TDDEs have advanced rapidly. Physics-Informed Neural Networks (PINNs) approximate solutions with neural networks whose loss enforces the governing equations, boundary conditions, and data (Raissi et al., 2019; Karniadakis et al., 2021). Despite their flexibility, PINNs often exhibit training instability and reduced accuracy for long-time dynamics, and are sensitive to sampling, loss weighting, and architecture choices (Wang et al., 2021; Huang & Agarwal, 2023). Operator-learning approaches, such as Fourier Neural Operators (FNOs) (Li et al., 2021) and DeepONets (Lu et al., 2019; 2021a), directly learn mappings from initial conditions and parameters to spatiotemporal solutions, enabling fast evaluation and improved generalization. However, their accuracy typically deteriorates when extrapolating to settings beyond the training range. Other work treats TDDE solving as general spatiotemporal sequence modeling (Huang et al., 2025; Liu et al., 2024; 2025g), using convolutional encoder-decoders (Ronneberger et al., 2015) for spatial encoding and recurrent units (Yu et al., 2019) for temporal evolution. These surrogates provide low-cost inference but degrade when rolled out autoregressively over long horizons (Huang et al., 2025). Overall, ML solvers generally remain discrete-time: they advance solutions via explicit integration or autoregressive rollout, which limits temporal resolution and makes it difficult for them to meet the required accuracy and efficiency requirements.

## 5.3. Exploration of DSMs

Dynamical systems machines (DSMs) form an emerging computational paradigm known for high efficiency, massive parallelism, and the ability to exploit intrinsic physical dynamics for computation. A representative example is the Ising machine, a physical realization of the Ising model that has shown strong performance on NP-hard binary optimization tasks, often surpassing digital solvers in speed and energy efficiency. Ising machines have been applied to MAX-CUT (Wang & Roychowdhury, 2019; Böhm et al., 2019; Mohseni et al., 2022; Liu et al., 2025e;f; Cılasun et al., 2025), SAT (Sharma et al., 2023a;b; Jagielski et al., 2023; Su et al., 2023; Sun et al., 2025), and wireless communication (Singh et al., 2022; Sreedhara et al., 2023). Beyond binary optimization, recent work has extended DSMs to ML tasks in both binary and real-valued domains (Niazi et al., 2024; Liu et al., 2025d; Wu et al., 2024; Liu et al., 2025b; Song et al., 2024; Liu et al., 2025a;c), demonstrating their potential as efficient computational substrates for broader workloads. Most existing methods, however, are built around equilibrium behavior: they encode the desired solution as an equilibrium state and read out outputs after the system converges. As a result, they ignore the rich transient dynamics of the system, which naturally encodes a full trajectory. Consequently, existing DSMs underutilize a central advantage of this paradigm: their ability to process and represent information throughout the entire dynamical evolution.

## 6. Conclusion

This work introduces DS-TS, a novel accurate and efficient TDDE solver that exploits the continuous evolution of DSMs. DS-TS integrates three key components: Excitatory-Inhibitory Inspired Coupling, State-aware Dynamic Nonlinearity, and Hierarchical Temporal Integration. Together, these components enable DS-TS to capture the complex spatiotemporal dependencies inherent in TDDEs. Experimental results demonstrate that DS-TS achieves high-fidelity solutions while delivering orders-of-magnitude improvements in speed ($\sim 10^3\times$) and energy efficiency ($\sim 10^5\times$) compared to existing solvers.

## Acknowledgements

This work is supported by the U.S. Department of Energy, Office of Science, Office of Advanced Scientific Computing Research, for the DeCoDe project, in support of the MEER-CAT Microelectronics Science Research Center, under Contract DE-AC05-76RL01830. This work is also supported by DARPA under Contract W912CG25CA007, by NSF under Award No. 2610649 and No. 2326494, and by NERSC through DDR-ERCAP0035256.

## Impact Statement

This paper presents work whose goal is to advance the field of Machine Learning. There are many potential societal consequences of our work, none which we feel must be specifically highlighted here.

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

# A. Appendix

This appendix provides additional details, including (i) complete benchmark specifications with mathematical formulations in A.1, (ii) the implementation details of DS-TS in A.2, (iii) additional experimental results in A.3, and (iv) a discussion of potential limitations and future directions in A.4.

## A.1. Benchmark Specifications

This section presents detailed mathematical formulations, initial conditions, and boundary conditions for the five time-dependent differential equations (TDDEs) used in our evaluation.

### A.1.1. ADVECTION-DIFFUSION EQUATION

The advection-diffusion equation models the transport of a scalar quantity under a spatially varying drift field combined with diffusion terms:

$$\frac{\partial u}{\partial t} = \kappa \frac{\partial^2 u}{\partial x^2} - v(x) \frac{\partial u}{\partial x}, \tag{17}$$

where $u(x, t)$ is the transported quantity, $\kappa$ is the diffusivity, and $v(x)$ is the spatially varying velocity field. The equation is posed on the periodic interval $x \in [0, L)$ with $L = 1.0$. The diffusivity is $\kappa = 10^{-3}$, and the velocity field is given by $v(x) = x^2$. The initial condition is a Gaussian bump centered at $x = L/2$:

$$u(x, 0) = u_0(x) = \exp\left(-100 \left(x - \tfrac{1}{2}L\right)^2\right). \tag{18}$$

Spatial derivatives are discretized using second-order centered finite differences with periodic wrap-around, and time evolution is advanced using the explicit forward Euler scheme. Periodic boundary conditions are enforced so that material leaving one boundary re-enters from the opposite side, yielding a spatially continuous domain and preserving global mass conservation.

### A.1.2. REACTION-DIFFUSION EQUATION

The reaction-diffusion equation captures pattern-forming dynamics by coupling spatial diffusion with local logistic reaction kinetics, thereby modeling how transport and nonlinear growth together can generate structured spatiotemporal patterns:

$$\frac{\partial u}{\partial t} = \gamma \left(\frac{\partial^2 u}{\partial x^2} + \frac{\partial^2 u}{\partial y^2}\right) + r\, u(1 - u), \tag{19}$$

where $\gamma$ is the diffusion coefficient and $r$ is the reaction rate. The equation is posed on the periodic unit square $(x, y) \in [0, 1) \times [0, 1)$ with parameters $\gamma = 0.1$ and $r = 5.0$. The initial condition is a centered Gaussian bump:

$$u(x, y, 0) = \exp\left(-100 \left[(x - 0.5)^2 + (y - 0.5)^2\right]\right). \tag{20}$$

The Laplacian operator is discretized using second-order centered finite differences under periodic boundary conditions, and the solution is advanced in time using the explicit forward Euler method.

### A.1.3. WAVE EQUATION

The wave equation describes wave propagation with linear damping and external time-dependent forcing:

$$\frac{\partial^2 u}{\partial t^2} = c^2 \nabla^2 u - \beta \frac{\partial u}{\partial t} + s(x, y, t), \tag{21}$$

where $c$ is the wave speed, $\beta$ is the damping coefficient, and $s(x, y, t)$ is the external forcing term. The equation is posed on the periodic domain $(x, y) \in [0, 1) \times [0, 1)$ with parameters $c = 0.1$ and $\beta = 0.5$. We introduce the auxiliary variable $v = \partial u/\partial t$, yielding the first-order system:

$$\begin{aligned} \frac{\partial u}{\partial t} &= v, \\ \frac{\partial v}{\partial t} &= c^2 \nabla^2 u - \beta v + s(x, y, t). \end{aligned} \tag{22}$$

The initial condition is a Gaussian bump, and the initial velocity is zero:

$$u(x, y, 0) = \exp\left(-80\left[(x - 0.5)^2 + (y - 0.5)^2\right]\right), \qquad v(x, y, 0) = 0. \tag{23}$$

The external forcing is a standing sinusoidal wave:

$$s(x, y, t) = A \sin(2\pi k_x x) \sin(2\pi k_y y) \cos(2\pi f t), \tag{24}$$

with amplitude $A = 1$, wavenumbers $k_x = k_y = 1$, and temporal frequency $f = 1.0$. Spatial derivatives are computed using second-order centered finite differences with periodic wrap-around, and the coupled first-order system in Eq. (22) is advanced in time using the explicit forward Euler method.

### A.1.4. FOKKER-PLANCK EQUATION

The Fokker–Planck equation models the time evolution of a probability density under diffusion together with drift arising from a confining potential:

$$\frac{\partial u}{\partial t} = \nabla \cdot (\kappa \nabla u + u \nabla V), \tag{25}$$

where $\kappa$ is the diffusivity and $V(x, y)$ is the external potential. The equation is posed on the periodic domain $(x, y) \in [0, 1) \times [0, 1)$ with diffusivity $\kappa = 0.01$. The external potential is a quadratic well centered at $(0.5, 0.5)$:

$$V(x, y) = \frac{1}{2}\left[(x - 0.5)^2 + (y - 0.5)^2\right]. \tag{26}$$

The initial condition is constructed as an off-center Gaussian bump, subsequently normalized to satisfy the probability density constraint $\int u \, dx \, dy = 1$:

$$u(x, y, 0) = \frac{\tilde{u}(x, y)}{\int \tilde{u} \, dx \, dy}, \quad \text{where} \quad \tilde{u}(x, y) = \exp\left(-\left[(x - 0.3)^2 + (y - 0.7)^2\right]\right). \tag{27}$$

In the discrete setting, normalization is performed as $u_{i,j} \leftarrow u_{i,j} / \left(\sum_{i,j} u_{i,j} \Delta x \Delta y\right)$. The potential gradient $\nabla V$ is approximated using second-order centered finite differences with periodic wrap-around, and time integration is performed using the explicit forward Euler method.

### A.1.5. HYPERBOLIC EQUATION

This equation provides a higher-order hyperbolic benchmark with fourth-order temporal dynamics:

$$\frac{\partial^4 u}{\partial t^4} = \alpha \nabla^2 u + s(x, y), \tag{28}$$

where $\alpha$ controls the spatial diffusion strength and $s(x, y)$ is a time-independent forcing term. The equation is posed on the periodic domain $(x, y) \in [0, 1) \times [0, 1)$ with $\alpha = 0.01$. We introduce auxiliary variables:

$$v_1 = u, \quad v_2 = \frac{\partial u}{\partial t}, \quad v_3 = \frac{\partial^2 u}{\partial t^2}, \quad v_4 = \frac{\partial^3 u}{\partial t^3}, \tag{29}$$

yielding the first-order system:

$$\begin{aligned}
\frac{\partial v_1}{\partial t} &= v_2, \\
\frac{\partial v_2}{\partial t} &= v_3, \\
\frac{\partial v_3}{\partial t} &= v_4, \\
\frac{\partial v_4}{\partial t} &= \alpha \nabla^2 v_1 + s(x, y).
\end{aligned} \tag{30}$$

The initial condition is a Gaussian profile centered at $(1, 1)$, and all higher-order time derivatives are initialized to zero:

$$u(x, y, 0) = \exp\left(-\frac{(x - 1)^2 + (y - 1)^2}{\sigma^2}\right), \quad \sigma = 1, \tag{31}$$

$$\left.\frac{\partial u}{\partial t}\right|_{t=0} = \left.\frac{\partial^2 u}{\partial t^2}\right|_{t=0} = \left.\frac{\partial^3 u}{\partial t^3}\right|_{t=0} = 0. \tag{32}$$

The forcing is time-independent and given by

$$s(x, y) = A \sin(\pi x) \cos(\pi y), \qquad A = 1. \tag{33}$$

The Laplacian is approximated using second-order centered finite differences with periodic wrap-around, and the first-order system in Eq. (30) is advanced in time using the explicit forward Euler method.

### A.2. The Implementation Details of DS-TS

This section presents the implementation details of DS-TS, covering the construction of the training dataset, the preprocessing and organization of training examples, and the overall training procedure used to obtain the final dynamics.

---

**Algorithm 1** Training pipeline for DS-TS

---

**Require:** Dataset $\mathcal{D} = \{(\mathbf{x}^{(h)}, \mathbf{s}^{(h)})\}_{h=1}^H$; split fractions $(\rho_{\text{tr}}, \rho_{\text{val}}, \rho_{\text{te}}) = (0.7, 0.1, 0.2)$; batch size $B = 32$.
**Require:** Optimizer: Adam; learning rates $\eta_A = 10^{-3}$ (Stage A), $\eta_B = 5 \times 10^{-4}$ (Stage B), $\eta_C = 2 \times 10^{-4}$ (Stage C).
**Require:** Max epochs per stage $E_{\max} = 1000$; early-stopping patience $p = 100$.
 1: **Split data:** $(\mathcal{D}_{\text{tr}}, \mathcal{D}_{\text{val}}, \mathcal{D}_{\text{te}}) \leftarrow \text{SPLIT}(\mathcal{D}, \rho_{\text{tr}}, \rho_{\text{val}}, \rho_{\text{te}})$
 2: **Build loaders:** `train_loader`, `val_loader`, `test_loader`.
 3: **Initialize model:** DS-TS $f_\Theta$ with parameters $\Theta = \{W_B, W_E, W_I, W_C, \alpha\}$. Configure $\Theta$ for 8-bit integer QAT with fake-quantized weights and activations.
 4: **Loss:** training loss $\mathcal{L} = \text{MSE}(\hat{\mathbf{s}}, \mathbf{s})$; validation metric $m = \text{MAE}(\hat{\mathbf{s}}, \mathbf{s})$.
 5: **Stage A:** $\text{TRAINSTAGE}(f_\Theta, \{W_B, W_C\}, \eta_A, \texttt{train\_loader}, \texttt{val\_loader}, \texttt{ckpt\_A})$
 6: **Stage B:** $\text{TRAINSTAGE}(f_\Theta, \{W_E, W_I, \alpha\}, \eta_B, \texttt{train\_loader}, \texttt{val\_loader}, \texttt{ckpt\_B})$
 7: **Stage C:** $\text{TRAINSTAGE}(f_\Theta, \Theta, \eta_C, \texttt{train\_loader}, \texttt{val\_loader}, \texttt{ckpt\_C})$
 8: **Test:** load best checkpoint `ckpt_C`; report MAE on `test_loader`.

 9: **Subroutine:** $\text{TRAINSTAGE}(f_\Theta, \mathcal{T}, \eta, \texttt{train\_loader}, \texttt{val\_loader}, \texttt{save\_path})$
10:     $\text{SETTRAINABLE}(f_\Theta, \mathcal{T})$                                        // only params in $\mathcal{T}$ are updated
11:     $\text{opt} \leftarrow \text{Adam}(\mathcal{T}, \text{lr} = \eta)$
12:     $\text{best} \leftarrow +\infty$; `no_improve` $\leftarrow 0$
13:     **for** $e = 1$ **to** $E_{\max}$ **do**
14:         **Train:** set $f_\Theta$ to train mode
15:         **for** mini-batch $(\mathbf{x}, \mathbf{s})$ **in** `train_loader` **do**
16:             $\hat{\mathbf{s}} \leftarrow f_\Theta(\mathbf{x})$
17:             $\ell \leftarrow \text{MSE}(\hat{\mathbf{s}}, \mathbf{s})$
18:             `opt.zero_grad()`
19:             backpropagate $\ell$
20:             `opt.step()`
21:         **end for**
22:         **Validate:** set $f_\Theta$ to eval mode
23:         `val_mae` $\leftarrow \text{EVALUATEMAE}(f_\Theta, \texttt{val\_loader})$
24:         `sch.step(val_mae)`
25:     **if** `val_mae` $<$ best **then**
26:         best $\leftarrow$ `val_mae`; `no_improve` $\leftarrow 0$
27:         save checkpoint to `save_path`
28:     **else**
29:         `no_improve` $\leftarrow$ `no_improve` $+ 1$
30:         **if** `no_improve` $\geq p$ **then**
31:             **break**                                            // early stopping
32:         **end if**
33:     **end if**
34:     **end for**

---

*Figure 8.* MAE comparison on selected TDDEs in R1 (a) and R2 (b), contrasting DS-TS under ideal conditions (w/o noise) and under the noise model (w/ noise) with the lowest MAE achieved among all baselines.

### A.2.1. TRAINING DATA CONSTRUCTION

For each TDDE, we construct a supervised dataset $\mathcal{D} = \{(\mathbf{x}^{(h)}, \mathbf{s}^{(h)})\}_{h=1}^{H}$ by sampling discretized states $\mathbf{x}^{(h)} \in \mathbb{R}^N$ and evaluating the corresponding ground-truth spatial operator $\mathbf{s}^{(h)} = \mathcal{S}^{\star}(\mathbf{x}^{(h)})$ in the highest-order temporal layer. To handle $L$-th order TDDEs, we employ HTI, which converts the target TDDE into a first-order dynamical system by augmenting each spatial node with $L$ auxiliary temporal states. This forms a chain of integrators that allows DS-TS to match the temporal order of the target TDDE by activating the appropriate number of temporal states. With $N$ spatial grid points mapped to $N$ DS-TS nodes, the model parameterizes a learnable operator $\hat{\mathcal{S}}_{\Theta}(\mathbf{x})$ with trainable parameter set $\Theta = \{W_B, W_E, W_I, W_C, \alpha\}$:

- $W_B$ are the base mapping weights,

- $W_E$ and $W_I$ are the excitatory and inhibitory branch weights,

- $W_C$ is the polynomial coefficient generator,

- $\alpha$ is the residual gate scalar.

Training minimizes the mean squared error (MSE) between the predicted operators and their ground-truth counterparts over the training dataset, as defined in Eq. (11).

### A.2.2. THREE-STAGE TRAINING PROCEDURE

We employ a three-stage training strategy that decouples the optimization of different model components:

**Stage A: Backbone fitting.** We first freeze the excitatory and inhibitory (E/I) branch parameters and optimize only the base mapping $W_B$ and the polynomial coefficient generator $W_C$, with the goal of learning an accurate backbone approximation of the underlying operator.

**Stage B: Gating alignment.** Next, we freeze $W_B$ and $W_C$ and train only the E/I interaction weights $(W_E, W_I)$ together with the residual gate scalar $\alpha$, thereby calibrating the gating behavior and the associated interaction patterns.

**Stage C: Joint fine-tuning.** In the final stage, we unfreeze all parameters and perform end-to-end fine-tuning. This allows the model to refine all components jointly and improve overall accuracy and stability.

Algorithm 1 provides the pseudocode for the three-stage training procedure.

### A.3. Additional Experimental Results

In this section, we provide additional experimental results that complement the main-text evaluation. Fig. 8 reports MAE on selected TDDEs in R1 and R2, comparing the MAE of DS-TS under ideal conditions (w/o noise) and under the noise model (w/ noise) against the lowest MAE among all baselines in each configuration. Across both regions, incorporating noise leads to only a modest increase in MAE, and DS-TS preserves its advantage: even with noise, its MAE remains below that of the strongest baseline for every reported case. These results corroborate the robustness trend observed in Fig. 7 and demonstrate that DS-TS's accuracy gains persist under realistic hardware perturbations.

### A.4. Discussion of Potential Limitations and Future Directions

There are several directions that could further improve both the accuracy and practical efficiency of DS-TS. First, for governing equations that include high-order time derivatives, the current approach reformulates the problem as an equivalent

first-order system and integrates the resulting augmented state in time. While this is a standard and broadly effective strategy, future work could consider more advanced time-integration methods, such as adaptive step-size control, or structure-preserving integrators, to potentially enhance stability and accuracy.

Second, the present implementation assumes a fully coupled correlation structure among nodes, which is highly expressive but may be unnecessary for some systems where dependencies are effectively sparse. A promising direction is to introduce locality, sparsity, or physics-informed interaction patterns that tailor inter-node correlations and encourage structured connectivity. Such designs could improve accuracy while reducing the required number of resistors, thereby further increasing computational and hardware efficiency.

