# OpenReview forum: "Solving Time-Dependent Differential Equations with Physical Dynamical Systems"
_ICML.cc/2026/Conference — ICML 2026 spotlight_

### Official Review · Reviewer_bcKj · 2026-03-09

**Soundness:** 3
**Presentation:** 2
**Significance:** 3
**Originality:** 3
**Overall Recommendation:** 5
**Confidence:** 3

**Summary:**

This paper proposes a dynamical-system (DS) based “processor” (DS-TS) for solving time-dependent differential equations. The core idea is to generate solution trajectories in continuous time by aligning the target TDDE dynamics with the intrinsic dynamics of a learnable coupled ODE system. The model introduces (i) Laplacian-style interactions augmented with an excitatory/inhibitory coupling mechanism, (ii) a state-conditioned polynomial nonlinearity (dynamic coefficients), and (iii) a hierarchical temporal integration scheme that represents higher-order temporal derivatives via auxiliary states. The authors report strong accuracy on several benchmarks across multiple spatial resolutions and both short and long rollouts, and argue for large latency/energy advantages from a hardware implementation, supported by circuit-level simulations and a thermal-noise analysis.

**Compliance With Llm Reviewing Policy:**

Affirmed.

**Final Justification:**

All of my concerns have been addressed. In my opinion, this paper is worthy of acceptance at ICML.

**Key Questions For Authors:**

How would DS-TS handle non-autonomous TDDEs with explicit time dependence or forcing? Would you augment the state with time, add an input channel, or reconfigure weights over time?

**Limitations:**

yes

**Strengths And Weaknesses:**

**Strengths**
- The motivation is clear: time discretization is a bottleneck for real-time TDDE solving, and continuous-time physical dynamics is a reasonable direction to explore.
- The architecture is coherent. HTI targets higher-order time structure, SDN adds state-dependent nonlinearity, and E/I coupling provides an additional handle for richer spatial interactions and stability.
- The empirical results show consistently low MAE across multiple PDE families, multiple resolutions, and long rollouts, which suggests good stability.

**Weaknesses**
- The experimental setting and baseline comparisons are not fully clear or fully fair. Training seems to assume access to a ground-truth operator $S^{\star}(x)$ derived from the known PDE, while FNO/UNet baselines are typically trained from trajectory data without explicit PDE/operator access. This mismatch makes it hard to interpret the magnitude of the advantage.
- It is unclear how the approach handles non-autonomous systems (explicit time dependence or time-varying coefficients). The learning formulation in the main text is written as  $S^{\star}(x)$ without an explicit time input, so the mapping for general non-autonomous TDDEs needs clarification.
- The log-domain SDN description raises a concrete implementability question: terms like $ln(I_{IN})$ and $ln(c_{m,i})$ require strictly positive currents, but the algorithmic definitions do not obviously guarantee positivity. The paper should explain sign handling / zero handling explicitly.

---

> ### Author Rebuttal · Authors · 2026-03-31
>
> We sincerely appreciate the valuable and positive review. We address each concern below.
>
> ---
> > `W1 Baseline Comparison`
>
> Thanks for the insightful comment. We will clarify and strengthen the comparison.
>
> Our baselines span both operator-based and trajectory-based training paradigms. PINN explicitly accesses the governing TDDEs through the residual term in its loss, while FNO and UNet are trained from trajectory data with implicit TDDE access. DS-TS outperforms both categories, demonstrating its effectiveness.
>
> Importantly, DS-TS can also be trained in a trajectory-based manner. To provide a more comprehensive comparison, we conduct additional experiments where DS-TS is trained on trajectory data, denoted as DS-TS (T). In this setting, we get the spatial operator targets via finite differences, $\\mathcal{S}^{\\star}(x_i(t)) = (x_i(t+\\Delta t) - x_i(t))/\\Delta t$. The results are listed below. DS-TS (T) achieves accuracy on par with the original DS-TS and even exhibits slight improvements in several cases, further supporting the effectiveness of the proposed method.
>
> | R1 | AD (ST) | AD (LT) | W (ST) | W (LT) | RD (ST) | RD (LT) | FP (ST) | FP (LT) | H (ST) | H (LT) |
> | --- | --- | --- | --- | --- | --- | --- | --- | --- | --- | --- |
> | DS-TS | 1.03e-6 | 4.33e-6 | 1.71e-5 | 4.99e-4 | 4.15e-6 | 3.15e-5 | 2.23e-6 | 8.96e-6 | 1.54e-6 | 7.09e-6 |
> | DS-TS (T) | 1.00e-6 | 4.27e-6 | 1.68e-5 | 4.87e-4 | 4.11e-6 | 3.07e-5 | 2.20e-6 | 8.92e-6 | 1.57e-6 | 7.14e-6 |
>
> | R2 | AD (ST) | AD (LT) | W (ST) | W (LT) | RD (ST) | RD (LT) | FP (ST) | FP (LT) | H (ST) | H (LT) |
> | --- | --- | --- | --- | --- | --- | --- | --- | --- | --- | --- |
> | DS-TS | 1.37e-6 | 5.19e-6 | 1.74e-5 | 5.07e-4 | 7.12e-6 | 3.22e-5 | 3.03e-6 | 1.21e-5 | 1.25e-6 | 1.02e-5 |
> | DS-TS (T) | 1.31e-6 | 5.11e-6 | 1.78e-5 | 5.16e-4 | 7.19e-6 | 3.35e-5 | 3.01e-6 | 1.17e-5 | 1.29e-6 | 1.14e-5 |
>
> | R3 | AD (ST) | AD (LT) | W (ST) | W (LT) | RD (ST) | RD (LT) | FP (ST) | FP (LT) | H (ST) | H (LT) |
> | --- | --- | --- | --- | --- | --- | --- | --- | --- | --- | --- |
> | DS-TS | 1.91e-6 | 8.48e-6 | 1.76e-5 | 5.14e-4 | 8.17e-6 | 4.16e-5 | 3.64e-6 | 1.40e-5 | 2.57e-6 | 1.24e-5 |
> | DS-TS (T) | 1.84e-6 | 8.35e-6 | 1.74e-5 | 5.09e-4 | 8.13e-6 | 4.11e-5 | 3.68e-6 | 1.47e-5 | 2.59e-6 | 1.37e-5 |
>
> Here, AD, W, RD, FP, and H represent Advection-Diffusion, Wave, Reaction-Diffusion, Fokker-Planck, and Hyperbolic, respectively. ST and LT represent short-term and long-term scenarios, respectively.
>
> ---
> > `W2&Q1 On Non-autonomous Systems`
>
> We appreciate your valuable comment and will provide clearer explanations.
>
> 1. Our benchmarks include non-autonomous TDDEs. The damped Wave equation (Eq. 21) includes an explicit time-dependent forcing term $s(x,y,t)$. DS-TS achieves the best performance on this benchmark, demonstrating that it handles non-autonomous TDDEs effectively. The reason DS-TS works is that, for a system evolving from a given initial state, the trajectory $\\mathbf{x}(t)$ is actually an implicit function of time $t$. Consequently, though $\\hat{S_{\\Theta}}(\\mathbf{x})$ does not receive $t$ as an explicit input, it receives $\\mathbf{x}(t)$, which implicitly encodes $t$. The proposed SDN exploits this: its coefficients $c_{m,i}(t)$ are functions of the instantaneous system state, so as the system evolves, the SDN coefficients adapt, causing the effective nonlinear operator to vary along the trajectory. This mechanism endows DS-TS with an implicit time-encoding.
>
> 2. To quantify the effect of providing time as an explicit input for non-autonomous TDDEs, we conduct additional experiments on the Wave equation where DS-TS receives both the state $\\mathbf{x}$ and the time $t$ as inputs. Results are reported below, where DS-TS (w/ t) denotes DS-TS with explicit time input. As shown, DS-TS (w/ t) yields similar performance to the original DS-TS, supporting the effectiveness of the original design.
>
> | | Wave (R1-ST) | Wave (R1-LT) | Wave (R2-ST) | Wave (R2-LT) | Wave (R3-ST) | Wave (R3-LT) |
> | --- | --- | --- | --- | --- | --- | --- |
> | DS-TS | 1.71e-5 | 4.99e-4 | 1.74e-5 | 5.07e-4 | 1.76e-5 | 5.14e-4 |
> | DS-TS (w/ t) | 1.73e-5 | 4.97e-4 | 1.65e-5 | 5.11e-4 | 1.82e-5 | 5.27e-4 |
>
> ---
> > `W3 On the Log-domain SDN Implementation`
>
> We apologize for any confusion and will clarify it clearly. The concern stems from interpreting $ln()$ under mathematical conventions, which requires a positive argument. However, in a physical circuit, "sign" simply indicates the direction of current flow (into vs. out of a node), while the magnitude is always positive. To handle both directions, our circuit first detects the current's direction and routes it to one of two parallel branches. Each branch then independently performs the log-domain polynomial computation on the positive current magnitude. This is a standard technique enabled by the two complementary transistor types (NMOS and PMOS), each naturally suited to processing one current direction.

---

> > ### Author Rebuttal · Reviewer_bcKj · 2026-04-02
> >
> > I thank the authors for their clear response. Since all of my concerns have been addressed, I will raise my score.

---

> > > ### Author Response · Authors · 2026-04-03
> > >
> > > Dear Reviewer bcKj,
> > >
> > > We sincerely thank you for your continued time and effort in reviewing our manuscript. We greatly appreciate your careful re-evaluation and are encouraged to know that the revisions have addressed your concerns satisfactorily.
> > >
> > > Your comments have played an important role in strengthening the manuscript, and we are grateful for the thoughtful guidance you have provided throughout the review process. We sincerely thank you for your time, expertise, and your valuable input.
> > >
> > > Best regards,
> > >
> > > The authors

---

### Official Review · Reviewer_Bv1H · 2026-03-13

**Soundness:** 4
**Presentation:** 3
**Significance:** 3
**Originality:** 4
**Overall Recommendation:** 5
**Confidence:** 2

**Summary:**

The work proposes DS-TS, an extension on the existing Dynamical System Machines (DSMs) for accurately and efficiently solving time-dependent differential equations (TDDEs). DS-TS is inspired on a biological neural network and augments spatial interactions with excitatory and inhibitory dynamics to better represent complex spatial dependencies (EIC). In addition, the algorithm introduces State-aware Dynamic Nonlinearity (SDN) to capture higher-order, dynamic nonlinear spatial interactions. Because SDN is a function of the system's states, it allows for adjustments within the local nonlinearities within an evolving system. Lastly, DS-TS introduces Hierarchical Temporal Integration (HTI) to transform TDDEs with higher-order time derivatives to an equivalent first-order system, which is easier to solve. The accuracy and efficiency of DS-TS is supported by experimental findings on 4 TDDE benchmarks.

**Compliance With Llm Reviewing Policy:**

Affirmed.

**Final Justification:**

The paper is technically solid and the initial concerns that I have had were only minor. Therefore, I maintain my original recommendation.

**Key Questions For Authors:**

1. I did not fully understand the connection of Equation 10 with the right Pane in Figure 3. Could the authors therefore elaborate on the methodology behind HTI? I am both a little puzzled by the conceptual meaning of Equation 10, as well the notation with the $\odot$ operators.

**Limitations:**

The authors include a section on the limitations of the work in Appendix 4, but do not include a reference towards this section in the main text.

**Strengths And Weaknesses:**

## Soundness 4
The work is, as far as I can estimate, technically sound and well-supported by experimental findings. The setup is well-designed, carefully tackling the difficulties associated with the hardware implementation through Quantization-aware training (Section 3.2) and stabilizing the solutions through the explicit handling of boundary conditions that stem from the spatial nature of DSMs (Section 3.3).

## Presentation 3
The submission is well structured, and the overall narrative is easy to follow. The Introduction includes a clear problem setting, and the work is sufficiently grounded in related literature. In addition, the authors clearly highlight the differences in implicit or explicit integration for spatial and temporal discretization, however, the Section 5 could benefit from a more explicit explanation on how DS-TS differs from existing numerical or machine-learning based methods. In addition, I found the use of the $\odot$ operators in Equation7 and 10 unclear, but that could be due to my limited background. I really appreciate the pseudocode provided in Appendix A.2, as it improves my understanding of the method and enhances the reproducibility of the paper.


## Significance 3
The paper introduces a novel TTDE solver that is more accurate than the current-state of the art, while having a similar latency. Therefore, DS-TS forms a valuable contribution to dynamical process modeling. Yet, the application of the method, being an extension of DSMs, is rather domain-specific.


## Originality 4
With the introduction of, Excitatory-Inhibitory Inspired Coupling, State-aware Dynamic Nonlinearity and Hierarchical Temporal integration, the work provides a valuable extension to existing Dynamical System Machines. The work may benefit from a more clear distinction of the current contribution from the existing literature, in particular w.r.t. the integration approach employed by DS-TS, compared to numerical alternatives.

---

> ### Author Rebuttal · Authors · 2026-03-31
>
> We sincerely thank the reviewer for the positive and constructive review. We address each concern below.
>
> ---
> > `Differentiation from Existing Methods`
>
> Thanks for your valuable suggestion. We will expand Sec. 5 to clearly differentiate DS-TS from numerical and ML methods.
>
> DS-TS solves TDDEs through continuous-time physical evolution. Functioning as a digital twin of the TDDE solving process, DS-TS programs a dynamical system whose intrinsic evolution is aligned with the target TDDE, thereby directly generating the desired trajectory.
>
> In contrast, numerical solvers approximate TDDE solutions through explicit or implicit time-stepping. Their accuracy is coupled to the discretization step size, leading to a latency-accuracy trade-off. Instead, DS-TS is inherently continuous-time, thereby breaking this trade-off and achieving high accuracy. Since the computation of DS-TS is realized through a natural system evolution, it also offers high efficiency.
>
> ML methods accelerate computation by compressing the multi-step operations of numerical solvers into an ML forward pass. As a result, they still learn discrete-time mappings, such as next-step predictors (e.g., FNO/UNet) or surrogates evaluated at discrete points (e.g., PINNs), limiting temporal fidelity and long-horizon accuracy. By contrast, DS-TS represents a novel paradigm for solving TDDEs accurately and efficiently.
>
> ---
> > `Hadamard Product`
>
> We appreciate your valuable comment and will clarify it clearly. $\\odot$ denotes the element-wise (Hadamard) product. $(\\cdot)^{\\odot m}$ denotes element-wise exponentiation to the $m$-th power, e.g., if $\\mathbf{z}=[z_1,z_2]^{\\top}$, then $\\mathbf{z}^{\\odot 2}=[z_1^2,z_2^2]^{\\top}$.
>
> ---
> > `Method behind HTI`
>
> We appreciate your insightful question and will provide a clearer explanation.
>
> HTI is designed to address TDDEs with higher-order time derivatives, e.g., $\\partial^3 x/\\partial t^3$. HTI lifts the system dynamics to an augmented state space. For each spatial node $i$, the augmented state includes the solution and its time derivatives:
> - $x_i(t)$: the solution,
> - $x_i^1(t)$: the first order derivative $\\partial x_i(t)/\\partial t$,
> - $x_i^L(t)$: the highest-order derivative.
>
> These states are linked by Eq. 9 $d x_i^\\ell/dt = x_i^{\\ell+1}$, forming a chain of integrators from the highest-order state to lower-order ones, finally yielding solution $x_i(t)$.
>
> ---
> > `Eq. 10 and Its Connection with Fig. 3`
>
> We apologize for any confusion and will clarify it clearly. Eq. 10 defines the dynamics of the highest-order state $d x_i^L(t)/dt$ in HTI. Specifically, $(B_i(t)+\\alpha(E_i(t)-I_i(t)))$ is the signal produced by EIC. SDN then applies a dynamic polynomial mapping $\sum_{m=0}^{D} c_{m,i}(t) \\odot (\\cdot)^{\\odot m}$ to this signal. Together, EIC and SDN determine the dynamics $d x_i^L(t)/dt$. HTI then propagates this top-layer dynamics to the solution through Eq. 9 $d x_i^\\ell(t)/dt=x_i^{\\ell+1}(t)$.
>
> The right panel of Fig. 3 illustrates this process. Each column corresponds to one node with augmented temporal states $x_i(t),x_i^1(t),\\ldots,x_i^L(t)$, and each row corresponds to one temporal layer $\\ell$. The top row $L$ receives input from the EIC & SDN modules, corresponding to Eq. 10. The downward arrows between rows represent the integration chain in Eq. 9. The bottom row $\\ell=0$ is the solution state $x_i(t)$.
>
> For example, for a TDDE with a second-order time derivative $L=2$. Eq. 10 specifies $d x_i^2(t)/dt$. HTI realizes a continuous integration chain over time: the evolving state $x_i^2(t)$ serves as the input to the next layer, so that $d x_i^1(t)/dt=x_i^2(t)$; $d x_i(t)/dt=x_i^1(t)$. All layers evolve simultaneously in continuous time to get the final solution $x_i(t)$.
>
> ---
> > `The Significance of DS-TS`
>
> We appreciate the reviewer's recognition that DS-TS is a valuable contribution to dynamical process modeling. We will make our scope clearer.
>
> DS-TS is developed for solving TDDEs. Though it appears to be a specific domain, it is in fact highly fundamental and broadly relevant. Virtually every system that evolves over time can be formulated by TDDEs: fluid dynamics in climate modeling, epidemic spreading, cascading failures in power grids, and even the score-based diffusion processes underlying generative models. The analysis, control, and optimization of these systems are often based on solving TDDEs.
>
> In many domains, real-time and accurate TDDE solving remains a central challenge, as existing methods are often either too slow or insufficiently accurate. In this regard, DS-TS opens up new opportunities by making previously impractical or unattainable capabilities feasible.
>
> Beyond TDDE solving, DS-TS can also be extended to broader domains, such as spatiotemporal modeling, real-time digital twins, autonomous systems, and so on.
>
> ---
> > `Limitations Section Reference`
>
> Thanks for your valuable suggestion. We will add a reference to Appendix A.4 (limitations) in conclusion.

---

> > ### Author Rebuttal · Reviewer_Bv1H · 2026-04-01
> >
> > Thank you for the elaborate rebuttal. I appreciate the proposed expansions and clarifications. As far as I can judge, I think that the manuscript is in a good state.

---

> > > ### Author Response · Authors · 2026-04-01
> > >
> > > Dear Reviewer Bv1H,
> > >
> > > Thank you very much for taking the time to re-evaluate our manuscript and for your thoughtful and constructive feedback throughout the review process. We are very glad to know that your concerns have been adequately addressed and that you think the manuscript is now in good shape.
> > >
> > > Your insightful comments and suggestions have been invaluable in helping us improve the quality and clarity of our work. We sincerely appreciate your time, expertise, and constructive review.
> > >
> > > Best regards,
> > >
> > > The authors

---

### Official Review · Reviewer_9fzN · 2026-03-13

**Soundness:** 2
**Presentation:** 2
**Significance:** 2
**Originality:** 2
**Overall Recommendation:** 4
**Confidence:** 3

**Summary:**

This paper introduces DS-TS, a novel solver for Time-Dependent Differential Equations (TDDEs) that leverages Dynamical System Machines (DSMs). The authors propose three synergistic innovations: Excitatory-Inhibitory Inspired Coupling (EIC), State-aware Dynamic Non-linearity (SDN), and Hierarchical Temporal Integration (HTI) to overcome the limitations of existing DSMs. The work claims significant improvements in speed and energy efficiency compared to digital baselines and ML surrogates across five TDDE benchmarks.

**Compliance With Llm Reviewing Policy:**

Affirmed.

**Final Justification:**

My major concerns are addressed. I have raised my score to 4. I hope these revisions and additions will be reflected in the final version of the paper, and that the code will be made publicly available to help advance the field.

**Key Questions For Authors:**

- Can the authors clarify if the reported $10^3$× speedup includes the time required for the three-stage "Hardware-Aware Training"?
- How does the solver handle PDE parameters (e.g., Reynolds number) that fall outside the training distribution, given that SDN coefficients are state-dependent?
- Since the HTI uses a "programmable switch network," how much energy and latency does this switching/reconfiguration logic contribute to the total system overhead?

**Limitations:**

yes

**Strengths And Weaknesses:**

Strengths: The paper presents a highly original and technically creative architecture that successfully extends Dynamical System Machines (DSMs) from static optimization to transient, high-order TDDE solving, demonstrating significant potential for ultra-efficient real-time scientific computing.

Weaknesses:
- Lack of Full-System Empirical Validation: The most critical weakness is that the results appear to be derived from a "mixed evaluation flow" rather than a full physical chip. While SDN behavior is based on hardware measurements, the global state evolution is simulated in software. Projections of $10^3$× speedup and $10^5$× energy efficiency without a fabricated full-system measurement are overly optimistic and lack the empirical rigor expected for such hardware-centric claims.
- Training Bottleneck: The three-stage training process (Algorithm 1) relies on ground-truth operators $S^*$ generated by traditional solvers. This suggests that for every new PDE or parameter set, a high-fidelity digital simulation must first be performed, potentially negating the efficiency gains of the DSM during the deployment phase.
- Hardware Scalability and Noise: While the authors provide a noise analysis (Eq. 16), physical DSMs often suffer from non-ideal effects like thermal drift and calibration errors in resistors/capacitors that accumulate over time. The "hierarchical temporal integration" might exacerbate these errors, but the discussion on long-term stability in a truly analog environment is insufficient.

---

> ### Author Rebuttal · Authors · 2026-03-31
>
> We sincerely appreciate your thoughtful and constructive review. We address each concern below and will add new explanations and results to our manuscript.
>
> ---
> > `W1 Physical Chip Verification`
>
> We appreciate this valuable concern. Our simulation-based evaluation follows established academic and industry standards. It uses finite element analysis with a 10 ps timestep and models standard hardware cells. Since DS-TS is built entirely from standard CMOS components with well characterized and validated behavior, its simulation accuracy is well supported.
>
> We agree that real-chip implementation may introduce overheads, e.g., power delivery inefficiencies, but they typically cause \~5× degradation [1]. Since our gains are around $10^3$ in speed and $10^5$ in energy efficiency, the advantages remain significant.
>
> ---
> > `W2 Training Requires Digital Simulation`
>
> We appreciate your insightful comment and apologize for any confusion.
> 1. DS-TS training does not require digital simulation. DS-TS training target $\mathcal{S}^{\star}$ is a pointwise spatial operator, derived from the governing equation, and its inputs $\mathbf{x}$ can be sampled from an estimated state range, without trajectory simulation.
> 2. In real-world applications such as digital twins, power grids, and nuclear fusion, state measurements are routinely collected and maintained, allowing DS-TS to be trained on readily available data.
>
> ---
> > `W3 Hardware Non-idealities`
>
> We appreciate this important comment. Our evaluation already accounts for circuit non-idealities following standard practices. The non-idealities are typically categorized into two types:
> - dynamic non-idealities, such as stochastic noise and random variations, for which Sec. 4.4 shows that DS-TS demonstrates strong robustness;
> - static non-idealities, such as thermal drift and systematic calibration errors, which are mitigated through standard techniques such as periodic offset compensation and reference calibration. The associated overhead is included in our reported results.
>
> ---
> > `Q1 If The Speedup Includes Training?`
>
> Thanks for your valuable question. The reported speedup refers to inference-time latency. We would like to clarify that:
> - DS-TS training is efficient. Instead of learning full trajectory rollouts, it trains only the operator at the highest-order temporal layer, yielding a compact model and 20× faster training than ML baselines.
> - Compared to numerical solvers that suffer from high solving latency, DS-TS's solving latency is negligible, and its total cost (training + solving) remains comparable to that of numerical solvers. Moreover, DS-TS training cost is one-time, whereas numerical solvers incur full solving cost on every call.
> - The value of DS-TS goes beyond mere speedup. By reducing latency, DS-TS makes previously impractical capabilities feasible, opening up new opportunities, such as millisecond-scale power-grid operation and sub-microsecond nuclear fusion control.
>
> ---
> > `Q2 Handling Parameters Outside the Training Distribution`
>
> We appreciate this insightful question. To evaluate DS-TS generalization, we generate TDDE trajectories under multiple parameter settings and randomly splitting them into train/validation/test sets (70%/10%/20%), ensuring that test parameters are unseen during training. We compare DS-TS against the strongest baseline, FNO (AR-16). The average test MAEs for the ST & R1 scenarios are shown below, with more results at https://anonymous.4open.science/r/material-/table.pdf. DS-TS consistently maintains low MAE on unseen parameters, demonstrating strong generalization.
>
> |  | Advection-Diffusion | Wave | Reaction-Diffusion | Fokker-Planck | Hyperbolic |
> | --- | --- | --- | --- | --- | --- |
> | FNO (AR-16) | 2.46e-4 | 3.35e-4 | 2.17e-4 | 3.38e-4 | 2.15e-4 |
> | DS-TS | 1.15e-6 | 2.64e-5 | 4.06e-6 | 3.32e-6 | 1.59e-6 |
>
> To further assess generalization in practical applications, we introduce a real-world power grid benchmark governed by the Swing equation, where the test set parameters unseen during training. The test MAEs below show that DS-TS generalizes effectively to unseen parameters.
>
> |  | Power Grid |
> | --- | --- |
> | FNO (AR-16) | 3.72e-4 |
> | DS-TS | 6.19e-5 |
>
> ---
> > `Q3 Overhead of the Programmable Switch Network`
>
> We appreciate this valuable comment. Quantitatively, the switch network overhead accounts for less than 0.001% of the total latency and less than 0.2% of the total power.
> - Latency: With an equivalent transistor resistance of \~100 Ω and a capacitance of \~20 fF, the bandwidth is 1/(2πRC) ≈ 79 GHz, far exceeding our signal bandwidth (~1 MHz), introducing less than 0.001% delay.
> - Power: From $I^2R$, with R ≈ 100 Ω and current ranging from 1–50 μA, the per-switch power is on the order of $10^{-10}$ to $2.5\times10^{-7}$ W. Even with all switches on, it takes less than 0.2% of total system power.
>
> ---
> [1] Ambrogio et al., An analog-AI chip for energy-efficient speech recognition and transcription, Nature, 2023.

---

> > ### Author Rebuttal · Reviewer_9fzN · 2026-04-04
> >
> > Thanks for your detailed rebuttle, I will raise my score.

---

> > > ### Author Response · Authors · 2026-04-06
> > >
> > > Dear Reviewer 9fzN,
> > >
> > > Thank you for your kind response and for agreeing to raise the score. We are very glad to know that your concerns have been adequately addressed.
> > >
> > > We just wanted to gently follow up, as it appears that the updated score has not yet been reflected in the system. We would greatly appreciate it if the score could be updated accordingly.
> > >
> > > Please also feel free to let us know if there are any additional questions or concerns. We would be more than happy to address them.
> > >
> > > Your thoughtful comments and guidance throughout the review process have been invaluable in improving the manuscript. We truly appreciate your time, expertise, and your constructive input.
> > >
> > > Best regards,
> > >
> > > The authors

---

### Decision · Program_Chairs · 2026-04-30

**Decision:**

Accept (spotlight)

**Comment:**

This paper investigates Dynamical System Machines for solving time-dependent differential equations. In particular, several new ideas are proposed to address the issues of insufficient temporal integration and the inability to handle dynamic spatial correlations. During the initial review, some concerns were raised regarding, for example, the experimental setup and the readability of the paper; however, all of these concerns have been fully addressed. The reviewers acknowledge that the contributions of this paper are solid, and the paper should be accepted.